# Morphological and Phylogenetic Evidence Reveal Five New Telamonioid Species of *Cortinarius* (*Agaricales*) from East Asia

**DOI:** 10.3390/jof8030257

**Published:** 2022-03-02

**Authors:** Meng-Le Xie, Chayanard Phukhamsakda, Tie-Zheng Wei, Ji-Peng Li, Ke Wang, Yang Wang, Rui-Qing Ji, Yu Li

**Affiliations:** 1Engineering Research Center of Chinese Ministry of Education for Edible and Medicinal Fungi, Jilin Agricultural University, Changchun 130118, China; xiemengle1993@126.com (M.-L.X.); chayanard91@gmail.com (C.P.); lijipengfungi@163.com (J.-P.L.); lesireyang@163.com (Y.W.); 2Life Science College, Northeast Normal University, Changchun 130024, China; 3State Key Laboratory of Mycology, Institute of Microbiology, Chinese Academy of Sciences, Beijing 100101, China; weitiezheng@163.com (T.-Z.W.); wangk@im.ac.cn (K.W.); 4College of Plant Protection, Shenyang Agricultural University, Shenyang 110866, China

**Keywords:** *Basidiomycota*, China, ectomycorrhizal fungi, FESEM, multi-gene phylogeny, new taxa, taxonomy

## Abstract

Five new *Cortinarius* species, *C. neobalaustinus*, *C. pseudocamphoratus*, *C. subnymphatus*, *C. wuliangshanensis* and *C. yanjiensis* spp. nov., are proposed based on a combination of morphological and molecular evidence. *Cortinarius neobalaustinus* is characterized by a very weakly hygrophanous and yellowish-brown to brown pileus and small and weakly verrucose basidiospores. *Cortinarius pseudocamphoratus* can be characterized by a viscid pileus, a strongly unpleasant smell, amygdaloid to somewhat ellipsoid basidiospores and lageniform to subfusiform cheilocystidia. *Cortinarius subnymphatus* is identified by a strongly hygrophanous pileus that is reddish-brown with a black-brown umbo, a yellowish universal veil and ellipsoid to subamygdaloid basidiospores. *Cortinarius wuliangshanensis* is characterized by a moderately to strongly hygrophanous, translucently striated and yellowish to reddish-brown pileus and rather weakly and moderately verrucose basidiospores. *Cortinarius yanjiensis* is distinguished by a weakly to moderately hygrophanous and yellowish to brown pileus and moderately to rather strongly verrucose basidiospores. The phylogenetic analyses were performed with maximum likelihood and Bayesian inference methods based on the data set of nuc rDNA ITS1-5.8S-ITS2 (ITS), D1–D2 domains of nuc 28S rDNA (28S) and RNA polymerase II second largest subunit (*rpb2*), and the results show that *C. neobalaustinus*, *C. wulianghsanensis* and *C. yanjiensis* cluster in sect. *Illumini*, *C. pseudocamporatus* belongs to sect. *Camphorati* and *C. subnymphatus* belongs to sect. *Laeti*. In addition, a study of basidiospores under field emission scanning electron microscopy (FESEM) was conducted. An identification key for the five new species and related species from China is also provided.

## 1. Introduction

*Cortinarius* (Pers.) Gray is an important ectomycorrhizal agaric genus in *Agaricales* and few species have sequestrate basidiome morphologies [1,2,3]. Although *Cortinarius* is known to be the largest basidiomycete genus with more than 3000 species worldwide [4], there are many new species described every year [5,6,7]. This could be due to the application of a polyphasic approach that combines phylogenetic and morphological methods [8,9,10].

*Telamonia* (Fr.) Trog is a traditional *Cortinarius* subgenus characterized by a dry to more or less hygrophanous pileus and stipe, often without bright colours [3,11]. However, some species have a more or less bluish tinge and/or a brightly coloured universal veil [3,11]. Some phylogenetic studies have shown that this subgenus is polyphyletic, while many lower ranking taxa formed well-supported monophyletic clades [3,12,13,14]. The subgenus is divided into subg. *Telamonia* s.s. and other telamonioid sections, in which subg. *Telamonia* s.s. is a monophyletic entity containing the vast majority of species, and another group is a polyphyletic assembly of several sections of *Camphorati* (Liimat., Niskanen & Ammirati) Soop, B. Oertel & Dima, *Illumini* (Liimat., Niskanen & Kytöv.) Soop, B. Oertel & Dima, *Laeti* Melot, *Obtusi* Melot, etc. [14].

In this study, five undescribed telamonioid species of *Cortinarius* were collected from China. Morphological characteristics and molecular phylogenetic analyses supported the recognition of five new species within *Cortinarius* and confirmed their infrageneric taxonomic status.

## 2. Materials and Methods

### 2.1. Specimens and Morphological Description

The studied specimens were deposited in the Herbarium of Mycology, Jilin Agricultural University (HMJAU), and the Fungarium of the Institute of Microbiology, Chinese Academy of Sciences (HMAS). Macroscopic characteristics were based on field notes and photos captured in the field. Micromorphological data were obtained from the dried specimens, which were observed in 5% potassium hydroxide water solution and/or Melzer’s reagent under a light microscope, following Xie et al. [15]. The ornamentation of basidiospores was studied using a Hitachi, model SU8010, field emission scanning electron microscope (FESEM) in Jilin Agricultural University. Basidiospore measurements, averages based on 30–40 basidiospores per collection, and measurements in parentheses are exceptional. Factor Q is the value of the length divided by width.

### 2.2. Molecular Phylogeny

Total DNA was extracted from dried specimens, using a NuClean PlantGen DNA Kit (CWBIO, Taizhou, China). Primers ITS1F and ITS4 were used to amplify the nrDNA ITS1-5.8S-ITS2 (ITS) region [16,17]. The D1–D2 domains of nuc 28S rDNA (28S) were amplified with primers LR0R and LR7 [18]. The RNA polymerase II second largest subunit (*rpb2*) were amplified with primers bRPB2-6F and bRPB2-7.1R [19,20]. The PCR procedures are as follows: for the ITS region, initial denaturation at 95 °C for 5 min, followed by 35 cycles at 95 °C for 30 s, 48 °C for 30 s and 72 °C for 1 min, and a final extension of 72 °C for 10 min; for 28S, initial denaturation at 94 °C for 2 min, followed by 35 cycles at 94 °C for 30 s, 51 °C for 30 s and 72 °C for 1 min, and a final extension of 72 °C for 10 min; for *rpb2*, initial denaturation at 95 °C for 5 min, followed by 35 cycles at 95 °C for 30 s, 58 °C for 40 s and 72 °C for 1 min, and a final extension of 72 °C for 8 min. Sequencing was performed by Sangon Biotech (Shanghai, China) Co., Ltd. All newly generated sequences were deposited in GenBank (Table 1).

The taxon sampling strategy for the selection of sequences for phylogenetic trees was to choose related taxa based on a BLASTn search in GenBank within *Cortinarius* and based on Soop et al. [14]. Three partition datasets (ITS, 28S, *rpb2*) were separately aligned and manually adjusted with BioEdit 7.1.3.0 [21]. Phyutility 2.2 was used to concatenate the aligned datasets [22]. *Cortinarius cyanites* Fr. and *C. boreicyanites* Kytöv., Liimat., Niskanen & A.F.S. Taylor were selected as an outgroup for phylogenetic analyses of the combined dataset, following Xie et al. [7].

For phylogenetic analyses, Bayesian inference (BI) and maximum likelihood (ML) methods were used. For BI analysis, the best-fit model for each partition was estimated using the Akaike information criterion (AIC), implemented in MrModeltest 2.3 [23]. The BI analysis was performed with MrBayes 3.2.6 [24]. Four Markov chains were run for 2 runs from random starting trees for 500,000 generations, sampling every 100th generation. The first 25% of trees were discarded to build the 50% majority rule consensus tree. RAxML 8.2.12, implemented in raxmlGUI, was used for ML analysis, with a rapid bootstrapping algorithm of 1000 replicates [25,26]. All default parameters with the GTRGAMMA model were used in the ML analysis. For BI analysis, GTR + I + G, GTR + I + G and GTR + I were the best-fit model for ITS, 28S and *rpb2* partitions, respectively.

## 3. Results

### 3.1. Molecular Phylogeny

The ITS + 28S + *rpb2* dataset for phylogenetic analyses included 68 samples, representing 50 species. The resulting alignments were deposited at TreeBASE (http://www.treebase.org; submission ID S29164; accessed on 28 December 2021). The BI and ML trees showed similar topologies, and the ML tree was selected as the representative phylogeny (Figure 1).

The phylogenetic tree recovered four sections and one subgenus, which clustered into three clades, subg. *Telamonia* s.s., telamonioid and outgroup, respectively. The telamonioid clade consisted of sect. *Camphorati*, sect. *Illumini* and sect. *Leati*. *Cortinarius neobalaustinus*, *C. wuliangshanensis* and *C. yanjiensis* clustered in sect. *Illumini* (BPP/ML = 1.00/99%), in which *C. neobalaustinus* is sister to *C. balaustinus* Fr. (BPP/ML = 1.00/100%). *Cortinarius pseudocamphoratus* belonged to sect. *Camphorati* (BPP/ML = 1.00/100%) and formed a sister relationship (BPP/ML = 1.00/80%) with the clade of *C. camphoratus* (Fr.) Fr. and *C. putorius* Niskanen, Liimat. & Ammirati (BPP/ML = 0.98/85%). *Cortinarius subnymphatus* was retrieved as a sister species of C. *nymphatus* Kytöv., Niskanen, Liimat. & Bojantchev (BPP/ML = 1.00/99%) in sect. *Leati* (BPP/ML = 1.00/100%).

### 3.2. Taxonomy

*Cortinarius neobalaustinus* M.L. Xie, T.Z. Wei & Y. Li, sp. nov. Figure 2A1,A2 and Figure 3A1–A3.

Mycobank: MB 842322.

Diagnosis: Basidiomata small to medium-sized. Pileus weakly hygrophanous, innately fibrillose, whitish, yellowish to brown. Lamellae moderately crowded, greyish-yellow to yellowish-brown. Stipe subcylindrical to clavate or bulbous, white at first, then yellowish at upper part, basal mycelium white. Universal veil white, rather copious. Basidiospores small, on average 5.9–6.3 μm × 4.9–5.4 μm, subglobose to ovoid or broadly ellipsoid, finely verrucose.

Holotype: China. Jilin Province, Yanji County, Sandaowan Town, in *Quercus mongolica* forests, alt. 580 m, 43°16′ N, 129°7′ E, 8 September 2018, *Mengle Xie*, HMJAU58951.

Etymology: The name refers to the affinity to *Cortinarius balaustinus*.

Description: Pileus 20–50 mm, convex at first, then becoming plano-convex with a broad umbo; weakly hygrophanous, innately fibrillose; whitish to lightly yellowish-white at first, then yellowish-brown to brown. Lamellae emarginated, moderately crowded, light yellow to greyish-yellow at first, then yellowish-brown to brown, edge paler, uneven. Stipe 43–57 mm long, 9–12 mm thick on top, 11–18 mm thick at the base, subcylindrical to clavate or bulbous, white fibrils, white at first, somewhat yellowish-brown at the upper part, basal mycelium white. Universal veil white, rather copious. Context thick, white at first, later brownish. Odour: radish.

Basidiospores 5.6–6.8(–7.6) μm × 4.6–5.8(–6.8) μm, av. 5.9–6.3 μm × 4.9–5.4 μm, Q = 1.04–1.37, av. Q = 1.18–1.24, subglobose to broadly ellipsoid, finely verrucose, indextrinoid. Basidia clavate, four-spored. Lamellar edges fertile, with clavate cells. Lamellar trama hyphae lightly olivaceous-brown to olivaceous-brown, up to 33 μm wide, finely encrusted. Pileipellis duplex: epicuit hyphae colourless to lightly olivaceous-brown, 2–6.5 μm wide, smooth. Hypodermium developed, hyphae lightly olivaceous-brown, 7–25 μm wide, smooth. Clamp connections present.

Ecology and distribution: Gregarious in *Quercus mongolica* forests in northeast of China and mixed forests of *Picea* sp. and *Quercus semicarpifolia* in southwest of China. Known from Jilin Province, Tibet Autonomous Region and Yunnan Province of China.

Additional specimens examined: China. Jilin Province, Yanji County, Sandaowan Town, in *Quercus mongolica* forest, alt. 580 m, 43°16′ N, 129°7′ E, 3 September 2017, Mengle Xie, HMJAU58948, HMJAU58949, HMJAU58950; ibid., 4 September 2019, Mengle Xie, HMJAU58952. Tibet Autonomous Region, Nyingchi City, Lulang Town, mixed forest with *Picea* sp. and *Quercus semicarpifolia*, 7 September 2014, Tiezheng Wei, Jianyun Zhuang, Xiaoyong Liu & Hao Huang, HMAS271994; ibid., 13 September 2014, Tiezheng Wei, Jianyun Zhuang, Xiaoyong Liu & Hao Huang, HMAS272377; ibid., 22 September 2015, Tiezheng Wei & Binbin Li, HMAS275263, HMAS254451, HMAS254466. Yunnan Province, Shangri-La County, Bitahai Nature Reserve, mixed forest with *Picea* sp. and *Quercus semicarpifolia*, 12 August 2008, Tiezheng Wei, HMAS250504.

Notes: *Cortinarius neobalaustinus* is characterized by the weakly hygrophanous yellowish-brown to brown pileus, small-sized basidiospores with finely verrucose and finely encrusted hyphae of the lamellar trama. It is a typical member of sect. *Illumini*. *Cortinarius balaustinus* shows similarity to the new taxon for its brown pileus and small basidiospores, but the basidiospores of *C. balaustinus* are moderately verrucose [27,28]. In addition, *C. balaustinus* is a widely distributed species in the Northern Hemisphere, usually under *Betula*, but less often with *Carpinus*, *Corylus* and *Quercus* trees [27,28]. Molecularly, (in ITS) based on the BLASTn, the most closely related species is *C. balaustinus*, which differs by 20 substitutions and indel positions, with a similarity of 96.5%. Table 2 provides the critical characteristics distinguishing new species and their similar species in sect. *Illumini*.

*Cortinarius pseudocamphoratus* M.L. Xie, T.Z. Wei & Y. Li, sp. nov. Figure 2B1,B2 and Figure 3B1–B3.

Mycobank: MB 842324.

Diagnosis: Pileus hemispherical to plano-convex, viscid, not hygrophanous, greyish-white when young with slightly violet tinge. Lamellae pale violet then bluish-brown, edge paler. Stipe cylindrical to clavate, violet. Universal veil whitish. Basidiospores on average 9.8–9.9 μm × 6.3–6.4 μm, amygdaloid to somewhat ellipsoid, finely and densely verrucose. Cheilocystidia present, somewhat lageniform to subfusiform.

Holotype: China. Tibet Autonomous Region, Linzhi City, Sejila Mountain, in *Abies* forest with *Rhododendron* spp., alt. 4200 m, 29°36′56″ N, 94°41′54″ E, 30 August 2019, *Mengle Xie*, HMJAU48694.

Etymology: The name refers to the affinitive species *Cortinarius camphoratus*.

Description: Pileus 26–55 mm diam., hemispherical, later plano-convex, viscid, not hygrophanous; greyish-white when young, with slightly violet tinge, later yellow; surface with veil remnants, especially on the margin. Lamellae emarginated, moderately crowded, violet when young, later bluish-brown to brown, edge paler, uneven. Stipe 50–75 mm long, 6–15 mm at above, 9–28 mm at the base, cylindrical to clavate, violet, especially at the apex, coated by abundant veil layer, more violet tinge visible when scraped. Universal veil whitish, then yellowish, very copious, forming incomplete girdles on the stipe. Context of the pileus whitish-yellowish, violet at the apex of the stipe, gradually transitioning to yellowish-grey at the base. Odour strong and unpleasant, typical of *C. camphoratus*.

Basidiospores 8.7–11.6 μm × 5.8–7 μm, av. 9.8–9.9 μm × 6.3–6.4 μm, Q = 1.35–1.90, av. Q = 1.50–1.60, amygdaloid to somewhat ellipsoid, finely and densely verrucose. Basidia clavate, four spores, colourless or lightly olivaceous-brown to olivaceous-brown. Lamellar trama hyphae lightly olivaceous-brown, up to 25 μm wide, smooth. Lamellar edge fertile, with small clavate sterile cells. Cheilocystidia present, somewhat lageniform to subfusiform, 28–53 μm × 9–13 μm, colourless. Pileipellis duplex: epicuits in weakly gelatinous substance, hyphae lightly olivaceous-brown to olivaceous-brown, 3–7 μm wide, smooth. Hypodermium developed, hyphae colourless to lightly olivaceous-brown, 5–25 μm wide, smooth. Trama hyphae colourless to lightly olivaceous-brown, smooth. Clamp connections present.

Ecology and distribution: Solitary or gregarious on moist soil of *Abies* forest with *Rhododendron* spp. Known from Qinghai–Tibetan Plateau of China.

Additional specimens examined: China. Tibet Autonomous Region, Linzhi City, Sejila Mountain, in *Abies* forests with *Rhododendron* spp., alt. 4200 m, 29°36′56″ N, 94°41′54″ E, 30 August 2019, Mengle Xie, HMJAU48698; 5 September 2020, Mengle Xie, HMJAU48794, HMJAU48795, HMJAU48796, HMJAU48797, HMJAU48798.

Notes: *Cortinarius pseudocamphoratus* is characterized by whitish with slightly violet-tinged basidiomata, viscid pileus, strong and unpleasant odour, amygdaloid to somewhat ellipsoid basidiospores, finely and densely verrucose and somewhat lageniform to subfusiform cheilocystidia, which correspond well to the circumscription of the sect. *Camphorati*. There are only five described species that belong to sect. *Camphorati* [14,30]. Three species are only distributed in Australasia, whereby *C. dysodes* and *C. tasmacamphoratus* are associated with *Nothofagus* and *C. austrotorvus* is a *Eucalyptus*-associated species. Another two species, *C. camphoratus* and *C. putorius*, are conifer-associated species and distributed in the Northern Hemisphere. However, *C. camphoratus* has a more violet and non-viscid pileus and the universal veil is lilac when young [28,31,32]. *Cortinarius putorius*, meanwhile, usually has a thin universal veil and the basidiospores are smaller, 8.8–9.5(–10) μm × 5–5.7 μm [30]. Phylogenetically, *C. pseudocamphoratus* cluster in sect. *Camphorati* and form a sister relationship with the clade of *C. camphoratus* and *C. putorius*. Molecularly, (in ITS) based on the BLASTn, the most closely related species is *C. camphoratus*, which differs by 35 substitutions and indel positions, with a similarity of 94.1%. Table 3 provides the critical characteristics distinguishing *C. pseudocamphoratus* and its similar species in sect. *Camphorati*.

*Cortinarius subnymphatus* M.L. Xie, T.Z. Wei & Y. Li, sp. nov. Figure 2C and Figure 3C1–C3.

Mycobank: MB 842325.

Diagnosis. Pileus conical to plano-convex with an umbo reddish-brown, black-brown at the centre, yellowish at the margin, strongly hygrophanous. Lamellae rusty brown. Stipe cylindrical. Universal veil yellowish. Basidiospores on average 7.6–7.7 μm × 5.0–5.1 μm, ellipsoid to subamygdaloid, finely to moderately verrucose, lamellar trama hyphae encrusted.

Holotype. China. Inner Mongolia Autonomous Region, Genhe County, Mangui Town, in mixed forests of *Larix gmelinii*, *Betula platyphylla* and *Vaccinium*, alt. 630 m, 52°03′52″ N, 122°04′40″ E, 24 August 2017, Mengle Xie, HMJAU48633.

Etymology. The name refers to the affinity to *Cortinarius nymphatus*.

Description: Pileus 24–52 mm diameter, somewhat conical at first, later convex to somewhat planar and umbonate; strongly hygrophanous; reddish-brown to dark reddish-brown, black-brown at the center, yellowish at the margin; very slightly translucently striated at the margin. Lamellae adnexed to emarginated, moderately crowded, rusty brown, sometimes with dark spots, edge uneven. Stipe 44–67 mm long, 5–8 mm thick, cylindrical; with white fibrillose. Basal mycelium white. Universal veil fairly distinct, yellowish, forming many thin patches and incomplete girdles on the stipe. Context fairly thin, strongly hygrophanous and brownish when moist. Odour indistinct.

Basidiospores 7.2–8.0(–8.7) μm × 4.6–5.4 μm, av. 7.6–7.7 μm × 5.0–5.1 μm, Q = 1.32–1.76, av. Q = 1.50–1.55, ellipsoid to subamygdaloid, finely to moderately verrucose. Basidia clavate, four spores, colourless to olivaceous-brown. Lamellar trama hyphae lightly olivaceous-brown to olivaceous-brown, up to 25 μm wide, moderately encrusted. Lamellar edge fertile, with clavate sterile cells. Pileipellis duplex: epicuit hyphae 5–17 μm wide, olivaceous-brown, zebra-striped, encrusted to finely encrusted. Hypodermium hyphae lightly olivaceous-brown, 3–30 μm wide, zebra-striped, encrusted to almost smooth. Trama hyphae lightly olivaceous to olivaceous, encrusted. Clamp connections present.

Ecology and distribution. Gregarious in a mixed forest of *Larix gmelinii*, *Betula platyphylla* and *Vaccinium*. Known from northeast China.

Additional specimens examined. China. Inner Mongolia Autonomous Region, Genhe County, Mangui Town, in mixed forest of *Larix gmelinii*, *Betula platyphylla* and *Vaccinium*, alt. 630m, 52°03′52″ N, 122°04′40″ E, 24 August 2017, Mengle Xie, HMJAU48632.

Notes: *Cortinarius subnymphatus* is characterized by a reddish-brown, mat and strongly hygrophanous pileus with a black-brown umbo, yellowish universal veil, ellipsoid to subamygdaloid basidiospores and encrusted hyphae of the lamellar trama. Morphologically, *C. subnymphatus* is similar to *C. fulvescens* and *C. fulvescentoideus*. *Cortinarius fulvescens* and *C. fulvescentoideus*, however, usually have bigger basidiospores, over 8 μm long [33]. Phylogenetically, *C. subnymphatus* clustered in sect. *Laeti*, sister to *C. nymphatus*. However, the pileus of *C. nymphatus* is redder, without the black-brown umbo, the basidiospores are usually less than 5 μm wide and associated with *Pinus* and *Picea* [33]. Molecularly, (in ITS) based on the BLASTn, the most closely related species is *C. nymphatus*, which differs by 10 substitutions and indel positions, with a similarity of 98.4%. Table 4 provides the critical characteristics distinguishing *C. subnymphatus* and its similar species in sect. *Laeti*.

*Cortinarius wuliangshanensis* M.L. Xie, T.Z. Wei & Y. Li, sp. nov. Figure 2D1,D2 and Figure 3D1–D3.

Mycobank: MB 842326.

Diagnosis: Basidiomata small to medium-sized. Pileus hygrophanous, translucently striate, innately fibrillose, yellowish, then reddish-brown at the centre. Lamellae subdistant, greyish-yellow to yellowish-brown. Stipe bulbous at the base when young, then clavate, yellowish-white to yellow, yellowish-brown, basal mycelium white. Universal veil white. Basidiospores small, on average 5.1–5.7 μm × 4.1–4.6 μm, subglobose to broadly ellipsoid, rather finely verrucose.

Holotype: China. Yunnan Province, Jingdong Yi Autonomous County, Wuliangshan National Nature Reserve, in broadleaf forest of *Castanopsis*, *Myrica*, *Quercus*, *Schima* and Ericaceae, alt. 1870 m, 24°28′33″ N, 100°43′26″ E, 12 September 2020, Mengle Xie, HMJAU58940.

Etymology: The name refers to the type location, Wuliangshan Mountains.

Description: Pileus 24–77 mm, hemispherical to convex at first, then becoming planar, slightly depressed at the centre, wavy and eroded at the margin in mature condition; surface medium hygrophanous, with a clear hygrophanous zone, slightly translucently striated and crenated at the margin, innately fibrillose, silvery-whitish when young, then strongly hygrophanous, with distinct translucent striations; yellowish-white to greyish-yellow at first, reddish-orange to reddish-brown at the centre with age. Lamellae emarginated, subdistant, greyish-yellow at first, then yellowish-brown to brown, edge even. Stipe 28–80 mm long, 4–12 mm thick at above, 6–25 mm thick at the base, bulbous at the base when young, then clavate, with white fibrils, whitish to lightly yellowish-white, yellow to yellowish-brown with age, basal mycelium white. Universal veil white, rather sparse. Context thin, yellowish-white when young, later yellowish-brown to brown at the stipe, hollow at the stipe when mature. Odour: radish.

Basidiospores 4.8–6.1 (–6.8) μm × 3.9–4.8 (–5.1) μm, av. 5.1–5.7 μm × 4.1–4.6 μm, Q = 1.09–1.41, av. Q = 1.23–1.26, subglobose to broadly ellipsoid, rare ellipsoid, rather finely verrucose, moderately dextrinoid. Basidia clavate, four-spored. Lamellar edges fertile, with narrow clavate cells. Lamellar trama hyphae lightly olivaceous-brown to olivaceous-brown, up to 18 μm wide, smooth. Pileipellis with a very thin epicuits, hyphae 3–7.5 μm wide, equal, colourless to lightly olivaceous-brown, smooth. Hypodermium developed, hyphae colourless to lightly olivaceous-brown, 6–20 μm wide, cylindrical to subcellular, smooth. Clamp connections present.

Ecology and distribution: Gregarious in broadleaf forests of *Castanopsis*, *Myrica*, *Quercus*, *Schima* and Ericaceae or mixed with *Pinus*. Known from Yunnan Province of China.

Additional specimens examined: Yunnan Province, Jingdong Yi Autonomous County, Wuliangshan National Nature Reserve, in broadleaf forest of *Castanopsis*, *Myrica*, *Quercus*, *Schima* and Ericaceae, alt. 1870 m, 24°28′33″ N, 100°43′26″ E, 12 September 2020, Mengle Xie, HMJAU58941; ibid., in broadleaf forests of *Castanopsis*, *Myrica*, *Quercus*, *Schima* and Ericaceae alt. 1850 m, 24°28′32″ N, 100°43′28″ E, 12 September 2020, Mengle Xie, HMJAU58942; ibid., in mixed forest of *Castanopsis*, *Myrica*, *Pinus*, *Quercus*, *Schima* and Ericaceae, alt. 1850 m, 24°28’31″ N, 100°43’28″ E, 12 September 2020, Mengle Xie, HMJAU58943.

Notes: *Cortinarius wuliangshanensis* is characterized by moderately to strongly hygrophanous, yellowish to reddish-brown and distinctly translucently striated pileus, bulbous at the stripe base when young, small-sized and rather finely verrucose basidiospores and smooth hyphae of lamellar trama. It is a typical member of sect. *Illumini*. Morphologically, it is similar to *C. microglobisporus*, sharing the yellowish pileus and white fibrillate stipe when young. However, the pileus of *C. microglobisporus* is without the distinct translucent striations and never reddish-brown and it is only distributed in the Mediterranean under *Quercus cerris* trees [29]. Molecularly, (in ITS) based on the BLASTn, the most closely related species is *C. microglobisporus*, which differs by 33 substitutions and indel positions, with a similarity of 94.8%. Table 2 provides the critical characteristics distinguishing *C. wuliangshanensis* and its similar species in sect. *Illumini*.

*Cortinarius yanjiensis* M.L. Xie, T.Z. Wei & Y. Li, sp. nov. Figure 2E and Figure 3E1–E3.

Mycobank: MB 842327.

Diagnosis: Basidiomata small to medium-sized. Pileus weakly to moderately hygrophanous, innately fibrillose, yellowish to brown. Lamellae subdistant to moderately crowded, greyish-yellow to yellowish-brown. Stipe cylindrical to slightly clavate, usually tapered at the base, white at first, then yellowish or yellowish spots at the upper part, basal mycelium white. Universal veil white, spares. Basidiospores small, on average 5.9–6.2 μm × 5–5.2 μm, subglobose to broadly ellipsoid, rather strongly verrucose.

Holotype: CHINA. Jilin Province, Yanji County, Sandaowan Town, in *Quercus mongolica* forests, alt. 580 m, 43°16′ N, 129°7′ E, 4 September 2019, Mengle Xie, HMJAU58947.

Etymology: The name refers to the type location, Yanji county.

Description: Pileus 20–70 mm, convex at first, then becoming plano-convex with a broad umbo; weakly to moderately hygrophanous, innately fibrillose; lightly yellow, yellowish-brown to brown, paler at the margin. Lamellae adnexed to subadnate, subdistant to moderately crowded, greyish-yellow at first, then yellowish-brown to brown, edge even at first, then uneven. Stipe 25–83 mm long, 4–14 mm thick at above, 7–23 mm thick at the base, cylindrical to slightly clavate, usually with slightly tapered base, white fibrils, white at first, somewhat yellow or yellowish spots at the upper part, basal mycelium white. Universal veil white, spares. Context rather thick, white at first, then yellowish to yellowish-brown at the stipe. Odour: radish.

Basidiospores 5.5–6.8 μm × 4.6–5.8 μm, av. 5.9–6.2 μm × 5–5.2 μm, Q = 1.04–1.34, av. Q = 1.17–1.2, subglobose to broadly ellipsoid, rather strongly verrucose, moderately dextrinoid. Basidia clavate, four-spored. Lamellar edges fertile, with clavate cells. Lamellar trama hyphae colourless to lightly olivaceous-brown, cylindrical to subcellular, up to 28 μm wide, smooth. Pileipellis duplex: epicuit hyphae olivaceous-brown, 2.5–10 μm wide, smooth. Hypodermium developed, hyphae lightly olivaceous-brown, 5–20 μm wide, smooth. Clamp connections present.

Ecology and distribution: Gregarious in *Quercus mongolica* forests. Known from Jilin Province of China.

Additional specimens examined: Jilin Province, Yanji County, Sandaowan Town, in *Quercus mongolica* forests, alt. 580 m, 43°16′ N, 129°7′ E, 7 September 2018, Mengle Xie, HMJAU58944; ibid., 4 September 2019, Mengle Xie, HMJAU58945, HMJAU58946.

Notes: *Cortinarius yanjiensis* is characterized by a weakly to moderately hygrophanous and yellowish to brown pileus, tapered base of the stipe, small-sized and rather strongly verrucose basidiospores, and smooth hyphae of the lamellar trama. It is a typical member of sect. *Illumini*. Morphologically, it is similar to *C. balaustinus* and *C. neobalaustinus*, sharing the weakly hygrophanous pileus. However, the lamellar trama hyphae of *C. balaustinus* and *C. neobalaustinus* are finely encrusted. The basidiospores of *C. balaustinus* are moderately verrucose [27,28] while those of *C. neobalaustinus* are finely verrucose. Molecularly, (in ITS) based on the BLASTn, the most closely related species is *C. illuminus*, which differs by 43 substitutions and indel positions, with a similarity of 93.1%. Table 2 provides the critical characteristics distinguishing *C. yanjiensis* and its similar species in sect. *Illumini*.
A key to the species in sections *Camphorati*, *Illumini* and *Laeti* from China1. Universal veil white………………………………………………………………………2
1. Universal veil bright colour…………………………………………………(sect. *Laeti*) 8
2. Basidiomata strongly unpleasant odour, pileus viscid, universal veil white, basidiospores amygdaloid to somewhat ellipsoid, finely and densely verrucose, cheilocystidia somewhat lageniform………………………………(sect. *Camphorati*) *C. pseudocamphoratus*
2. Basidiomata usually indistinct to more or less like radish without unpleasant odour, basidiospores globose, subglobose, to broadly ellipsoid…………………(sect. *Illumini*) 3
3. Pileus with distinctly translucent stripe…………………………………………………4
3. Pileus without distinctly translucent stripe………………………………………………5
4. Pileus vivid reddish-brown, basidiospores moderately verrucose, hyphae of the lamellar trama finely encrusted, usually occur in mixed forests of *Larix gmelinii*, *Betula platyphylla* and *Vaccinium*, distributed in northeast China…………………*C. khinganensis*
4. Pileus yellowish at first, then reddish-brown, stipe usually bulbous when young, basidiospores rather finely verrucose, hyphae of the lamellar trama smooth, occur in broadleaf forests of *Castanopsis*, *Myrica*, *Quercus*, *Schima* and Ericaceae, distributed in southwest China……………………………………………………………*C. wuliangshanensis*
5. Pileus usually reddish-brown, basidiospores moderately verrucose……………………6
5. Pileus usually yellowish to brown, almost without reddish-brown……………………7
6. Pileus often concentrically hygrophanous, with slightly translucent stripe at the margin, usually occur in coniferous forests, distributed in southwest China………*C. illuminus*
6. Pileus weakly hygrophanous, never translucently striated, usually occur in deciduous forests, distributed in northeast China……………………………………*C. balaustinus*
7. Pileus usually yellowish at first, stipe cylindrical to slightly clavate, usually with slightly tapered base, universal veil spares, basidiospores rather strongly verrucose, hyphae of the lamellar trama smooth……………………………………………*C. yanjiensis*
7. Pileus whitish to slightly yellowish-white at first, stipe subcylindrical to clavate or bulbous, universal veil rather copious, basidiospores finely verrucose, hyphae of the lamellar trama finely encrusted………………………………………………*C. neobalaustinus*
8. Basidiospores subglobose to broadly ellipsoid……………………………………………9
8. Basidiospores ellipsoid to subamygdaloid………………………………………………10
9. Pileus distinctly hygrophanous, universal veil reddish-brown to vinaceous-brown……………………………………………………………………………*C. badiovinaceus*
9. Pileus weakly hygrophanous, universal veil yellowish-brown……………*C. ochrophyllus*
10. Universal veil pink……………………………………………………*C. roseobulliardioides*
10. Universal veil yellowish to ochraceous…………………………………………………11
11. Pileus reddish-brown, black at the centre…………………………….*C. subnymphatus*
11. Pileus yellowish-brown to black-brown…………………………………*C. cadi-aguirrei*

## 4. Discussion

In this study, five species, *Cortinarius neobalaustinus*, *C. pseudocamphoratus*, *C. subnymphatus*, *C. wuliangshanensis* and *C. yanjiensis*, are described from China as new species based on macro-, micro- and ultra-characteristics and multi-gene phylogeny. Phylogenetic analyses showed that the five new species clustered in the telamonioid clade, outside *Telamonia* s.s., which is consistent with other studies [3,12,13,14].

Section *Camphorati* was described as a subgenus in a previous study [34] but was reduced to a section based on multi-gene phylogenetic analyses [14]. Section *Camphorati* has the typical characteristics of medium to large-sized basidiomata, with a blue to purple tinge when young, a strongly unpleasant odour, basidiospores that are amygdaloid to somewhat ellipsoid and cheilocystidia that are somewhat lageniform. There are five species in this section worldwide [14,29], and in the present study, *C. pseudocamphoratus* is considered to be the sixth species worldwide and the only confirmed species from China in this section.

Section *Illumini* was described as a subgenus based on a previous study [34] but was reclassified at section level based on multi-gene phylogenetic analyses [14]. Section *Illumini* has the typical characteristics of medium to large-sized basidiomata, a yellowish to brown or reddish-brown pileus, being more or less hygrophanous and innately fibrillose, its basidiospores globose, subglobose, ovoid to broadly ellipsoid and weakly to rather strongly verrucose. There are six species in this section worldwide, with one species described from China [14,35]. The present study confirms the placement of six species in sect. *Illumini*, including three new species from China.

Section *Laetii* species were previously included in subg. *Telamonia* or subg. *Hydrocybe* [36,37] and as a section within sect. *Fulvescentes* Melot, distinguished by the colour of the universal veil [3,33]. However, previous phylogenetic analyses showed that sect. *Fulvescentes* is a synonym of sect. *Laetii* [12,13,14,33]. Section *Laeti* has the typical characteristics of small to medium-sized basidiomata, a pileus that is more or less hygrophanous, its universal veil a bright colour, its basidiospores subglobose, ellipsoid to amygdaloid and weakly to moderately verrucose. In this study, we confirmed five species in sect. *Laeti* distributed in China, with the new species *C. subnymphatus*, sister to *C. nymphatus*, a European species [33].

Most of the *Cortinarius* species were originally described in Europe and North America, while little work has been done in Asia [38]. In recent years, we have been devoted to researching *Cortinarius* in China. The present study reports 12 telamonioid species, including five new species, confirming and clarifying the species component of sections *Camphorati*, *Illumini* and *Laeti* in China, enriching the diversity of *Cortinarius*.

## Figures and Tables

**Figure 1 jof-08-00257-f001:**
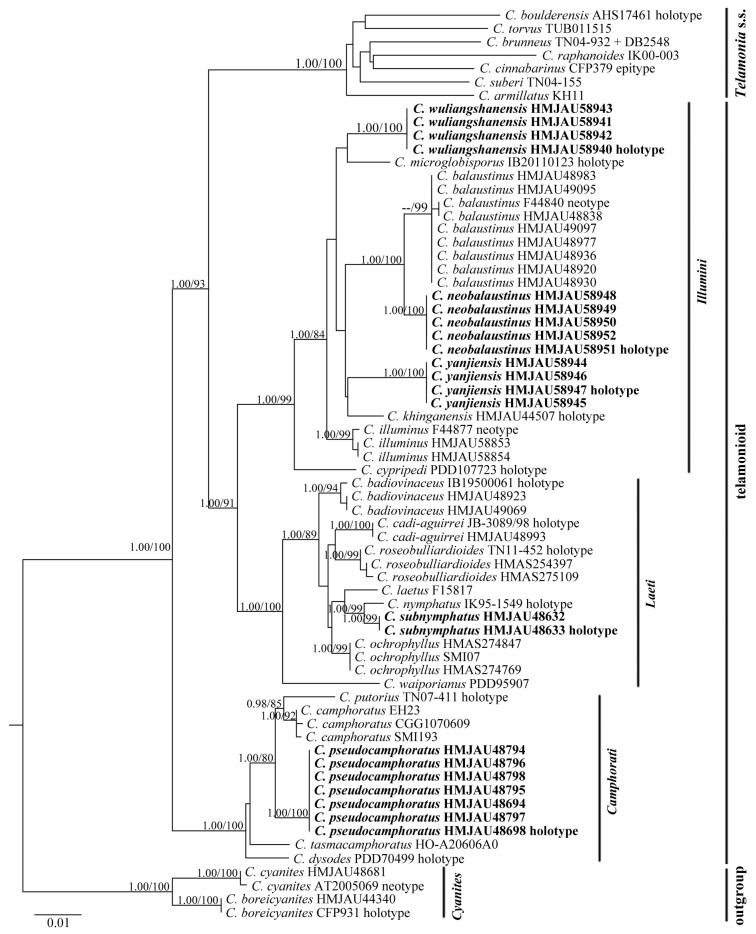
ML phylogram inferred from the ITS + 28S + *rpb2* dataset. The tree is rooted with sect. *Cyanites*. The Bayesian posterior probabilities (BPP) ≥ 0.95 and ML bootstrap values (ML) ≥ 75% are shown on the branches (BPP/ML). New species are marked in black bold font.

**Figure 2 jof-08-00257-f002:**
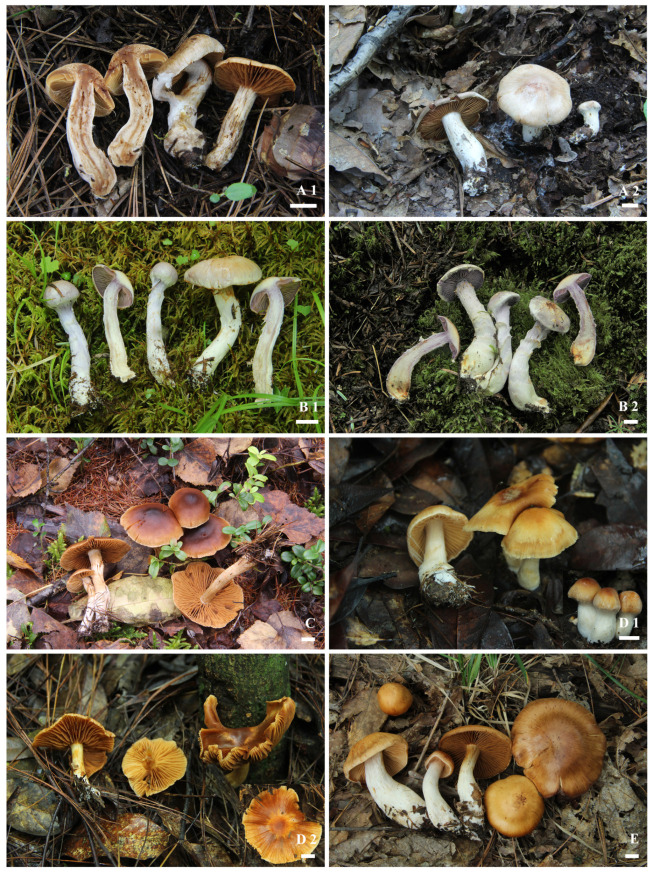
Basidiomata of *Cortinarius neobalaustinus*: (**A1**) HMJAU58951, holotype, (**A2**) HMJAU58948; *C. pseudocamphoratus*: (**B1**) HMJAU48698, holotype, (**B2**) HMJAU48795; *C. subnymphatus*: (**C**) HMJAU48633, holotype; *C. wuliangshanensis*: (**D1**) HMJAU58940, holotype, (**D2**) HMJAU58943; *C. yanjiensis*: (**E**) HMJAU58947, holotype. Bars = 10 mm.

**Figure 3 jof-08-00257-f003:**
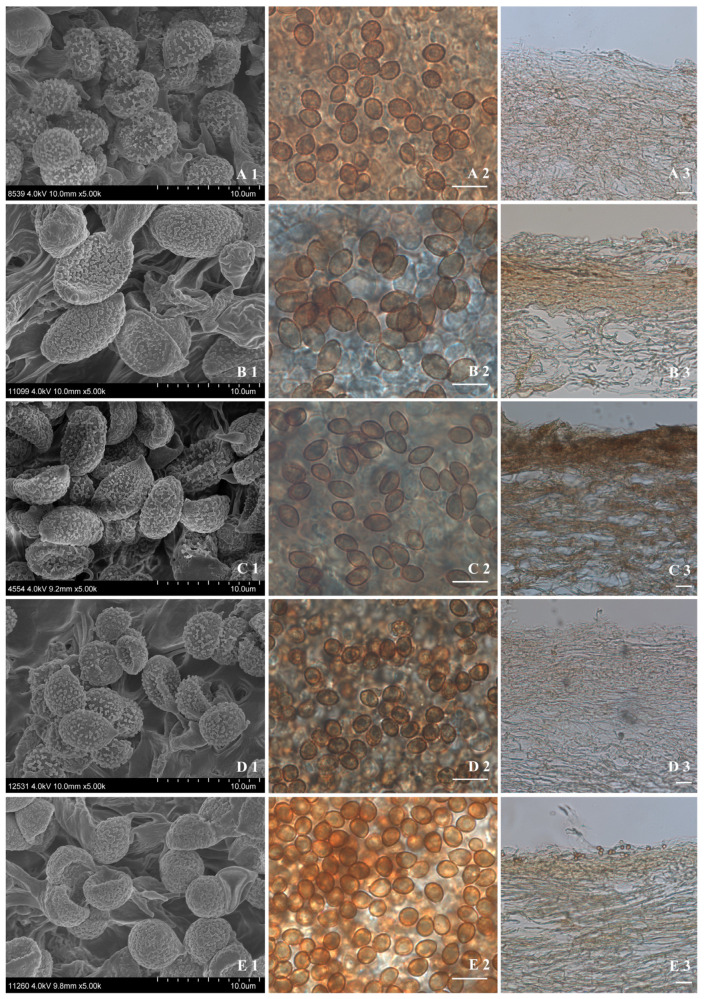
The basidiospores and pileipellis of the new species. (**A1**–**A3**) *C. neobalaustinus*, HMJAU58951, holotype; (**B1**–**B3**) *C. pseudocamphoratus*, HMJAU48698, holotype; (**C1**–**C3**) *C. subnymphatus*, HMJAU48633, holotype; (**D1**–**D3**) *C. wuliangshanensis*, HMJAU58940, holotype; (**E1**–**E3**) *C. yanjiensis*, HMJAU58947, holotype. (**A1**–**E1**) Basidiospores under FESEM; (**A2**–**E2**) Basidiospores under microscope, bars = 10 μm; (**A3**–**E3**) Pileipellis under microscope, bars = 20 μm.

**Table 1 jof-08-00257-t001:** Sequences used in the phylogenetic analyses.

Species	Voucher Number	GenBank Accession Number		
		ITS	28S	*rpb2*
*C. armillatus*	KH11	KC842408	—	KC842479
*C. badiovinaceus*	IB19500061 (holotype)	HQ845169	—	—
*C. badiovinaceus*	HMJAU48923	**OM001515**	—	—
*C. badiovinaceus*	HMJAU49069	**OM001516**	**OM001532**	**OM001816**
*C. balaustinus*	F44840 (neotype)	MW599265	—	—
*C. balaustinus*	HMJAU48838	**OM001501**	—	—
*C. balaustinus*	HMJAU48920	**OM001502**	—	—
*C. balaustinus*	HMJAU48930	**OM001503**	—	—
*C. balaustinus*	HMJAU48936	**OM001504**	—	—
*C. balaustinus*	HMJAU48977	**OM001505**	—	—
*C. balaustinus*	HMJAU48983	**OM001500**	**OM001529**	**OM001817**
*C. balaustinus*	HMJAU49095	**OM001506**	—	—
*C. balaustinus*	HMJAU49097	**OM001507**	—	—
*C. boreicyanites*	CFP931 (holotype)	KF732296	—	—
*C. boreicyanites*	HMJAU44340	**OM001482**	**OM001522**	**OM001811**
*C. boulderensis*	AHS17461 (holotype)	DQ499466	—	KC608614
*C. brunneus*	TN04-932	EU266638	—	JX407372
*C. brunneus*	DB2548	—	MK358063	—
*C. cadi-aguirrei*	JB-3089/98 (holotype)	KJ866953	—	—
*C. cadi-aguirrei*	HMJAU48993	**OM001521**	**OM001533**	**OM001818**
*C. camphoratus*	EH23	FJ717505	FJ717505	—
*C. camphoratus*	CGG1070609	MT775573	—	—
*C. camphoratus*	SMI193	FJ039626	FJ039626	—
*C. cinnabarinus*	CFP379 (epitype)	JX114944	—	KC608616
*C. cyanites*	AT2005069 (neotype)	KF732355	—	—
*C. cyanites*	HMJAU48681	MT299957	**OM001523**	**OM001812**
*C. cypripedi*	PDD107723 (holotype)	KT875199	KT875199	—
*C. dysodes*	PDD70499 (holotype)	GU233340	GU233394	
*C. illuminus*	F44877 (neotype)	KP866156	—	—
*C. illuminus*	HMJAU58853	**OM001490**	—	—
*C. illuminus*	HMJAU58854	**OM001491**	**OM001526**	—
*C. khinganensis*	HMJAU44507 (holotype)	MT299952	**OM001525**	**OM001814**
*C. laetus*	F15817	FJ157034	FJ157034	—
*C. microglobisporus*	IB20110123 (holotype)	NR153027	—	—
*C. neobalaustinus*	HMJAU58948	**OM001509**	—	—
*C. neobalaustinus*	HMJAU58949	**OM001510**	—	—
*C. neobalaustinus*	HMJAU58950	**OM001512**	—	—
*C. neobalaustinus*	HMJAU58951 (holotype)	**OM001508**	**OM001530**	**OM001819**
*C. neobalaustinus*	HMJAU58952	**OM001511**	—	—
*C. nymphatus*	IK95-1549 (holotype)	KX388639	—	—
*C. ochrophyllus*	HMAS274769	**OM001518**	—	—
*C. ochrophyllus*	HMAS274847	**OM001517**	—	—
*C. ochrophyllus*	SMI07	FJ039604	FJ039604	—
*C. pseudocamphoratus*	HMJAU48694	OM001484	—	—
*C. pseudocamphoratus*	HMJAU48698 (holotype)	**OM001483**	**OM001524**	**OM001813**
*C. pseudocamphoratus*	HMJAU48794	**OM001485**	—	—
*C. pseudocamphoratus*	HMJAU48795	**OM001486**	—	—
*C. pseudocamphoratus*	HMJAU48796	**OM001487**	—	—
*C. pseudocamphoratus*	HMJAU48797	**OM001488**	—	—
*C. pseudocamphoratus*	HMJAU48798	**OM001489**	—	—
*C. putorius*	TN07-411 (holotype)	KR011124	—	—
*C. raphanoides*	IK00-003	JX407333	—	JX407378
*C. roseobulliardioides*	TN11-452 (holotype)	KX388641	—	—
*C. roseobulliardioides*	HMAS254397	**OM001519**	—	—
*C. roseobulliardioides*	HMAS275019	**OM001520**	—	—
*C. suberi*	TN04-155	JX407336	—	JX407382
*C. subnymphatus*	HMJAU48632	**OM001514**	—	—
*C. subnymphatus*	HMJAU48633 (holotype)	**OM001513**	**OM001531**	**OM001820**
*C. tasmacamphoratus*	HO-A20606A0	AY669633	AY669633	—
*C. torvus*	TUB011515	AY669668	AY669668	—
*C. waiporianus*	PDD95907	MH101548	MH108387	MH141009
*C. wuliangshanensis*	HMJAU58940 (holotype)	**OM001496**	**OM001528**	**OM001821**
*C. wuliangshanensis*	HMJAU58941	**OM001497**	—	—
*C. wuliangshanensis*	HMJAU58942	**OM001498**	—	—
*C. wuliangshanensis*	HMJAU58943	**OM001499**	—	—
*C. yanjiensis*	HMJAU58944	**OM001492**	—	—
*C. yanjiensis*	HMJAU58945	**OM001493**	—	—
*C. yanjiensis*	HMJAU58946	**OM001494**	—	—
*C. yanjiensis*	HMJAU58947 (holotype)	**OM001495**	**OM001527**	**OM001815**

New sequences are shown in black bold font.

**Table 2 jof-08-00257-t002:** Morphological comparisons of the new species and their similar species in sect. *Illumini*.

Species	Pileus	Stipe	Spores	Hyphae of the Lamellar Trama
*Cortianrius balaustinus*	Weakly hygrophanous, yellowish-brown to reddish-brown, without translucently striate	Cylindrical to weakly clavate	Av. 6–6.1 μm × 4.9–5.1 μm, moderately verrucose	Finely encrusted
*C. illuminus*	Concentrically hygrophanous, reddish-brown, slightly translucently striate at the margin	Cylindrical to slightly clavate, sometimes with slightly tapered base	Av. 5.9 μm × 5.1 μm, moderately verrucose	Finely encrusted
*C. khinganensis*	Strongly hygrophanous, reddish-brown, distinctly translucently striate at the margin	Cylindrical to slightly clavate	Av. 6.8–6.9 μm × 5.9–6.0 μm, moderately verrucose	Finely encrusted
*C. microglobisporus* [29]	Hygrophanous zone, yellowish, slightly translucently striate at the margin	Cylindrical to slightly clavate or bulbous, sometimes with slightly tapered base	Av. 5.6 μm × 4.4 μm	—
*C. neobalaustinus*	Weakly hygrophanous, yellowish-brown to brown, without translucently striate	Subcylindrical to clavate or bulbous	Av. 5.9–6.3 μm × 4.9–5.4 μm, finely verrucose	Finely encrusted
*C. wuliangshanensis*	Strongly hygrophanous, yellowish to reddish-brown, distinct translucently striate	Clavate, somewhat bulbous when young	Av. 5.1–5.7 μm × 4.1–4.6 μm, rather finely to moderately verrucose	Smooth
*C. yanjiensis*	Weakly hygrophanous, yellowish-brown to brown, without translucently striate	Cylindrical to slightly clavate, usually with slightly tapered base	Av. 5.9–6.2 μm × 5–5.2 μm, rather strongly verrucose	Smooth

**Table 3 jof-08-00257-t003:** Morphological comparisons of the new species and their similar species in sect. *Camphorati*.

Species	Pileus	Stipe	Universal Veil	Spores
*Cortianrius camphoratus* [28,30,31,32]	Not hygrophanous, not viscid, blue to almost white, then yellowish-brown	Cylindrical to clavate	Lilac, then yellowish, very copious	(9–)9.5–10.5 μm × (5.5–)6–6.5 μm, finely and densely verrucose
*C. pseudocamphoratus*	Not hygrophanous, viscid, grayish-white, then yellowish	Clavate	Whitish, then yellowish, very copious	Av. 9.8–9.9 μm × 6.3–6.4 μm, finely and densely verrucose
*C. putorius* [30]	Not hygrophanous, viscid, purple, then pale purple to almost whitish	Cylindrical to somewhat clavate	White, sparse	Av. 9.2 μm × 5.4 μm, finely and densely verrucose

**Table 4 jof-08-00257-t004:** Morphological comparisons of the new species and their similar species in sect. *Laeti*.

Species	Pileus	Stipe	Universal Veil	Spores
*Cortinarius fulvescens* [33]	Strongly hygrophanous, reddish-brown to vinaceous reddish-brown	Cylindrical	Pink, very sparse	7.9–9.5 μm × 4.5–5.2 μm, often somewhat sharply verrucose
*C. fulvescentoideus* [33]	Strongly hygrophanous, warm reddish-brown	Cylindrical	Pale pink, very sparse	8.2–9.5 μm × (4.5–)5–5.4 μm, finely to moderately verrucose
*C. nymphatus* [33]	Strongly hygrophanous, brown to dark reddish-brown	Cylindrical	Yellow to ochraceous, forming incomplete girdles on the stipe	6.8–8.2 μm × 4.3–4.8 μm, finely to moderately, sharply verrucose
*C. subnymphatus*	Strongly hygrophanous; reddish-brown to dark reddish-brown, black-brown at the center	Cylindrical	Yellowish, forming many thin patches and incomplete girdles on the stipe	7.2–8.0(–8.7) μm × 4.6–5.4 μm, finely to moderately verrucose

## Data Availability

All resulting alignments were deposited in TreeBASE (http://www.treebase.org; accession number S29164; accessed on 28 December 2021). All newly generated sequences were deposited in GenBank (https://www.ncbi.nlm.nih.gov/genbank/; accessed on 23 December 2021). All new taxa were linked with MycoBank (https://www.mycobank.org/; accessed on 23 December 2021).

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
