# Peer review of "Morphological and Phylogenetic Evidence Reveal Five New Telamonioid Species of *Cortinarius* (*Agaricales*) from East Asia"

_jof, 2022, doi:10.3390/jof8030257_

Round 1

Reviewer 1 Report

The well-designed experimental part enables a very precise description and identificaton of five new telamonioid species of Cortinarius found in East Asia. Therefore, this research represents a significant contribution to a better understanding of the taxonomy of the genus Cortinarius.

Author Response

Dear reviewer,

Thank you for reviewing our manuscript.

Best wishes,

Meng-Le Xie

Reviewer 2 Report

Dear Editor/authors

I have reviewed the manuscript numbered JOF 1604972.
I consider that it is accepatble, however I consider that the discussions about the new species, in particular two of them that are marked in the text, should be more profound since it is not clear which are the concrete differences between the new species and the closest species.
At the same time I consider that some supports between the nodes of the phylogenetic analysis are somewhat low taking into account that 3 markers were used.
In the key it would be desirable to append nearby species even if they are not present in China in order to see the differences between them.

Specific observations were made in the attached pdf file

Author Response

Thank you for your carefully review of our article, which we have revised according to your comments. Please see the attachment

Best wishes,

Meng-Le Xie

Reviewer 3 Report

Overall, a nice job on this paper.  The quality of the photos of the basidiospores and sections of the pileipellis need to be improved. SEM photos are good.

Cortinarius is undergoing a lot of revision.  The species you are presenting are in Cortinarius sensu stricto.  Use subgeneric names to label your phylogeny, do not use the term "telamonioid".  Section Laetii is not being used currently in Cortinarius, use instead section Fulvescentes.  This section is not firmly plsced in a subgenus at this time but is part of Cortinarius sensu stricto.  I made notes on your manuscript for these points.

Be sure the descriptions are carefully written.

Author Response

(The authors gave the same response as above.)
